# REVISITING SUPERVISION FOR CONTINUAL REPRESENTATION LEARNING

## ABSTRACT

In the field of continual learning, models are designed to learn tasks one after the other. While most research has centered on supervised continual learning, there is a growing interest in unsupervised continual learning, which makes use of the vast amounts of unlabeled data. Recent studies have highlighted the strengths of unsupervised methods, particularly self-supervised learning, in providing robust representations. The improved transferability of those representations built with self-supervised methods is often associated with the role played by the multi-layer perceptron projector. In this work, we depart from this observation and reexamine the role of supervision in continual representation learning. We reckon that additional information, such as human annotations, should not deteriorate the quality of representations. Our findings show that supervised models when enhanced with a multi-layer perceptron head, can outperform self-supervised models in continual representation learning. This highlights the importance of the multi-layer perceptron projector in shaping feature transferability across a sequence of tasks in continual learning.

## 1 INTRODUCTION

In continual learning (CL), the goal of the model is to learn new tasks sequentially. Most of the works focus on supervised continual learning (SCL) for image classification where the learner is provided with labeled training data and the metric of interest is accuracy on all the tasks seen so far. More recently, unsupervised continual learning (UCL) gained more attention (Fini et al., 2022; Gomez-Villa et al., 2021; Madaan et al., 2022). UCL considers the problem of learning robust and general representations on a sequence of tasks, without accessing the data labels. Effective UCL methods would allow the utilization of vast amounts of unlabeled data emerging on a daily basis and continually improve existing models.

A number of recent works study continual learning from a representation learning perspective and show that unsupervised approaches build more robust representations when trained continually (Madaan et al., 2022; Davari et al., 2022). More specifically, Madaan et al. (2022) shows that self-supervised learning (SSL) methods build representations that are more robust to forgetting than supervised learning (SL). Davari et al. (2022) notices that training SimCLR (Chen et al., 2020) have advantageous properties for continual learning compared to supervised training. However, it is still counter-intuitive that access to more information (labels) results in worse representations in continual learning.

One of the potential reasons is the transferability gap between supervised and unsupervised learning. It was believed that the superior transferability of unsupervised learning can be attributed to a special design of contrastive loss (Zhao et al., 2020; Islam et al., 2021) or lack of annotations during training (Ericsson et al., 2020; Sariyildiz et al., 2020). However, recent works (Wang et al., 2021; Sariyildiz et al., 2023) identify that a multi-layer perceptron (MLP) projector commonly used in SSL (Chen et al., 2020; Chen & He, 2020; Zbontar et al., 2021; Grill et al., 2020) is a crucial component that improves transferability of SSL models. Following that founding Wang et al. (2021); Sariyildiz et al. (2023) use an MLP projector to improve transferability of supervised learning and achieve state-of-the-art transfer learning performance, surpassing unsupervised methods.

In this work, encouraged by these advancements in improving the transferability of supervised models, we revisit supervision for continual representation learning. We argue that additional infor-

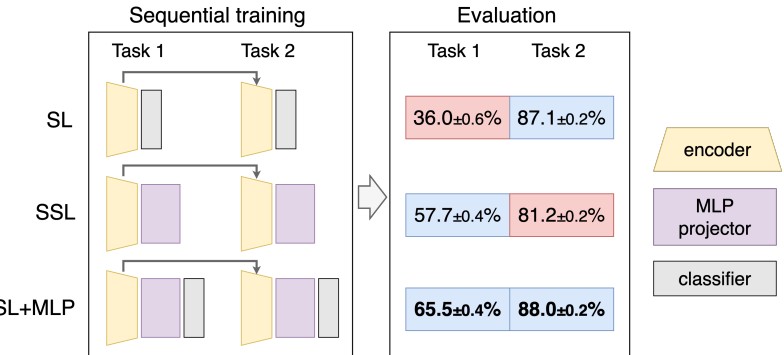

Figure 1: In a two-task continual learning scenario, supervised learning (SL) results in representations that perform well on the second task but poorly on the first task. On the other hand, representations trained with self-supervised learning (SSL) have higher first-task performance but they underperform on the second task. We show that simple modifications to supervised learning (SL+MLP) yield representations that are superior on the first task and on par with SL on the second task. We report average task-aware k-NN accuracy on 6 different scenarios (3 runs for each scenario).

mation (human annotations) should not hurt the quality of representations in continual learning, as suggested by Madaan et al. (2022). Motivated by the latest study on transferability of representations in self-supervised and supervised learning, we aim to improve transferability between tasks in continual learning. We are the first to show that supervised models can continually learn representations of higher quality than self-supervised models when trained with a simple MLP head (see Fig. 1). We identify the crucial role of an MLP projector in representation learning through the perspective of feature transferability, forgetting, and retention for continually trained models.

The main contributions of this paper are as follows:

- We empirically show that SL equipped with a simple MLP projector can learn higher-quality representations than SSL methods in continual finetuning scenarios.
- We show that the use of the MLP projector can be coupled with several continual learning methods, further improving their performance.
- We shed light on the reasons behind the strong performance of supervised learning with MLP projector: better transferability, lower forgetting, and increasing diversity of representations.

## 2 RELATED WORK

**Self-supervised Learning** Learning effective visual representations without annotations is a long-standing problem that aims at leveraging large volumes of unlabeled data. Recent SSL methods show impressive performance, matching or even exceeding the performance of their supervised equivalents (Chen et al., 2020; Grill et al., 2020; Zbontar et al., 2021; Caron et al., 2021; Chen & He, 2020). The majority of these techniques rely on image augmentation methods to produce multiple views for a given sample. They train a model to be insensitive to these augmentations by ensuring that the network generates similar representations for the positives. In this work, we use BarlowTwins (Zbontar et al., 2021) which considers an objective function measuring the cross-correlation matrix between the features and SimCLR (Chen et al., 2020) which uses contrastive learning based on noise-contrastive estimation. A number of studies (Bordes et al., 2023; Chen & He, 2020; Zbontar et al., 2021; Jing et al., 2022) show that an MLP projector between the encoder and the loss function is a crucial component to prevent the collapse of the representations and improve their transferability.

**Transferable representations** Wang et al. (2021) seeks to understand the transferability gap between unsupervised (SSL) and supervised pretraining. They found out that adding a projection network (which is commonly used in SSL) boosts the transferability of the supervised models' features. This was further explored in Sariyildiz et al. (2023) and it was shown that it is possible to

build representations that are good for both the source and the downstream tasks. In this work, we revisit those findings in the context of models learned on a sequence of tasks. Contrary to the transfer learning literature (Ericsson et al., 2020; Sariyildiz et al., 2023), which usually focuses on the downstream task performance, we evaluate the model on all tasks during the sequential training. This allows us to gain more insight into learned representations, i.e. representation forgetting.

**Supervised Continual Learning (SCL)** SCL aims to create systems that can acquire the ability to solve novel tasks using new annotated data while retaining the knowledge acquired from previously learned tasks (Parisi et al., 2019). A popular formulation of CL is class-incremental learning (CIL) (Masana et al., 2023; Van de Ven & Tolias, 2019) where each task introduces unseen classes that will not occur in the following tasks. In an exemplar-free setting, the model is not allowed to store any samples from previous tasks which might be important in situations where privacy concerns apply and such a setting remains a great challenge (Smith et al., 2022). A popular strategy is *feature distillation* (?Yan et al., 2021) which minimizes representational changes in subsequent learning stages by enforcing consistent output between the current model and the one trained in the previous task.

**Unsupervised Continual Learning (UCL)** Despite the success of SSL methods, they are designed to learn from large static datasets. UCL methods aim to overcome this issue and allow the models to learn from an ever-changing stream of data without excessive memory requirements. Recent works (Fini et al., 2022; Madaan et al., 2022; Gomez-Villa et al., 2021) apply SSL in the UCL setting and claim their superior results for continual representation learning. Most successful methods apply feature distillation through learnable non-linear projector: CaSSLe (Fini et al., 2022) distills features outputted by the projector while PFR (Gomez-Villa et al., 2021) distills the features outputted by the backbone. UCL models are evaluated by measuring their representation strength through linear probing or k-nearest neighbors (k-NN) and this paper follows this evaluation protocol.

## 3 EXPERIMENTAL SETUP

**Datasets** We utilize four different datasets: CIFAR10 (Krizhevsky, 2009) (C10), CIFAR100 (Krizhevsky, 2009) (C100), SVHN (Netzer et al., 2011) and ImageNet100 (Tian et al., 2019) (IN100), 100-class subset of the ILSVRC2012 dataset (Russakovsky et al., 2015) with $\approx$ 130k images in high resolution (resized to $224 \times 224$). We consider popular settings in continual learning: CIFAR10/5, CIFAR100/5, CIFAR100/20 and ImageNet100/5 sequences, where $D/N$ denotes that dataset $D$ is split into $N$ tasks with an equal number of classes in each task without overlapping ones. To gain further insight, we construct multiple two-task settings where we investigate representation strength and stability. We denote task shift with "$\rightarrow$", e.g. sequence $A \rightarrow B$ means that the model was trained on two tasks, the first one was dataset $A$ and the second one was dataset $B$. We consider C10$\rightarrow$C100 and C100$\rightarrow$C10 scenarios as having low distribution shifts, while C10$\rightarrow$SVHN and SVHN$\rightarrow$C10 scenarios involve higher distribution shifts.

**Methods** We use the following supervised methods: (1) SL - the standard approach of training a model with linear classification head with a cross-entropy loss function and a common set of data augmentations (resize, crop, flip) (Masana et al., 2023). (2) SL+MLP - SL with MLP projector added between the backbone and a linear head that is discarded at test-time and stronger data augmentations inspired by Wang et al. (2021) (more details are provided in Sec. 4.5). (3) t-ReX (Sariyildiz et al., 2023), and (4) SupCon (Khosla et al., 2020). Note that SL is the only method that does not utilize an additional MLP projector during training. For SSL approaches we choose BarlowTwins (Zbontar et al., 2021) and SimCLR (Chen et al., 2020). Results denoted as SSL were obtained using BarlowTwins. For CL strategies we use LwF (Li & Hoiem, 2018), CaSSLe (Fini et al., 2022) and PRF (Gomez-Villa et al., 2021) (details provided in Appendix A.1). We use ResNet-18 (He et al., 2016) as a feature extractor network for all the experiments.

**Training** We use the code repository from CaSSLe (Fini et al., 2022) and we follow their training procedure. We train SSL models for 500 epochs per task using SGD optimizer with momentum with batch size 256 and cosine learning rate schedule. We adapt the procedure to SL by reducing the number of epochs to 100 per task, the learning rate to 0.025, and the batch size to 64. We use augmentations from SimCLR (Chen et al., 2020) for SSL, augmentations proposed in Wang et al. (2021) for SL+MLP, and crop and horizontal flip for SL. Note that, unless stated otherwise, we

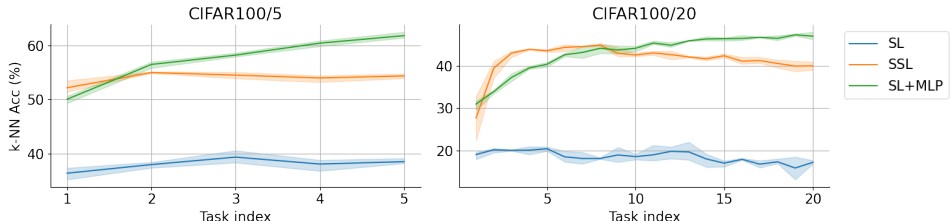

Figure 2: SL+MLP: (1) achieves strong performance after the initial task compared to SL which indicates that it produces representations that are transferable to the unseen tasks; (2) is the only method that is able to accumulate knowledge learned on a sequence of tasks. We report task-agnostic k-NN accuracy after each task on the whole dataset (notice that yet unseen tasks are also included in the evaluation).

investigate continual finetuning scenario and we do not employ any methods for continual learning nor replay buffer.

**Evaluation** We use k-NN classifier to evaluate the quality of representations following Fini et al. (2022); Madaan et al. (2022) and Nearest Mean Classifier (NMC) as in Rebuffi et al. (2017); Yu et al. (2020) to evaluate the stability of representations. We use CKA (Kornblith et al., 2019) to measure the similarity between representations of two models. Moreover, we use forgetting ($F$) and forward transfer ($FT$) commonly used in continual learning (Lopez-Paz & Ranzato, 2017). We also measure task exclusion difference $EXC$ (Hess et al., 2023) to evaluate the level of retention of task-specific features. We use subscripts to indicate the evaluation dataset, e.g. $Acc_{C10}$ means "accuracy on C10 dataset". We report means and standard deviations computed across 3 runs unless stated otherwise.

## 4 EXPERIMENTAL RESULTS

This section presents the experimental results of continual representation learning. In Section 4.1 we present our main results showing that supervised models can outperform self-supervised models in continual representation learning. The following sections shed light on the reasons for improved performance. Section 4.2 investigates the quality of representations, including forgetting, task exclusion comparison, similarity, and forward transfer. Section 4.3 presents a spectral analysis of representations. In Section 4.4, we evaluate the stability of the representations learned by different methods under various distribution shifts. Finally, in Section 4.5 we present an ablation study that shows the positive impact of MLP head and stronger augmentations on representations built in supervised continual learning.

### 4.1 MAIN RESULTS

Figure 2 presents our main results. Namely, we show that supervised models can build stronger representations than self-supervised models under continual finetuning, contrary to previous beliefs (Madaan et al., 2022). We identify that the key component to improving the performance of supervised models is an additional MLP projector used during training and discarded afterward - without it, SL significantly underperforms compared to SSL.

We identify two factors contributing to superior results of SL+MLP. Firstly, we observe that the performance of supervised models after the initial task is largely improved by the addition of the MLP projector, resulting in accuracy close to SSL models. In order to achieve good task-agnostic accuracy on the whole dataset (seen and unseen classes), the model trained on a single task needs to perform well on unseen data. Therefore, we attribute the advantage of SL+MLP to the increased transferability of representations induced by MLP projector, which is in line with Wang et al. (2021); Sariyildiz et al. (2023). Secondly, we notice that SL+MLP is the only method able to incrementally accumulate knowledge and consistently improve performance. This observation is in line with the increasing diversity of features presented in Sec. 4.3.

Table 1 presents extended results including multiple SL and SSL approaches in continual finetuning and paired with different CL methods.

Firstly, we observe that all the supervised methods equipped with the projector significantly outperform simple SL. SL+MLP, t-ReX, and SupCon achieve much higher results than SL in all the finetuning experiments. What is worth noting is the fact that all these methods were trained with different supervised losses: SL+MLP uses cross-entropy, t-ReX uses cosine softmax cross-entropy and SupCon uses supervised contrastive loss. However, they all utilize the MLP projector and all outperform SL.

Secondly, we show that most methods benefit from using CL strategies. Moreover, we observe that the positive effects of the MLP projector and CL strategy compound. As a result, the best models are those (1) trained in a supervised way (2) with the use of the MLP projector and (3) coupled with CL strategy based on temporal learnable projection, namely CaSSLe or PFR. The only exception is the CIFAR10/5 setting where SSL methods outperform SL. It is likely because each task in this scenario for SL is a binary classification task which does not encourage building meaningful representations.

| Method | CL strategy | C10/5 | C100/5 | C100/20 | IN100/5 |
|---|---|---|---|---|---|
| | | SUPERVISED CONTINUAL LEARNING | | | |
| SL | Finetune | 56.9±1.4 | 38.5±0.4 | 17.2±0.3 | 35.3±1.3 |
| | LWF | 62.2±1.1 | 57.4±0.2 | 45.2±1.2 | 60.5±0.3 |
| | PFR | 68.5±1.5 | 57.7±0.4 | 44.4±1.3 | 58.7±0.2 |
| SL+MLP | Finetune | 65.9±0.7 | 61.9±0.5 | 47.1±0.7 | 62.4±0.4 |
| | LWF | 72.6±3.4 | 58.7±0.2 | 51.9±0.1 | 60.4±0.2 |
| | PFR | 76.3±1.0 | **63.6±0.2** | **54.5±0.2** | 65.2±0.1 |
| t-ReX | Finetune | 69.3±1.1 | 59.2±0.6 | 50.8±0.1 | 59.2±0.6 |
| | LwF | 74.5±0.7 | 58.3±0.4 | 50.4±0.1 | 58.6±1.0 |
| | PFR | 75.9±1.2 | 60.9±0.5 | 53.4±0.3 | 63.9±0.6 |
| SupCon | Finetune | 60.4±0.6 | 49.4±0.3 | 30.0±0.7 | 57.6±0.6 |
| | CaSSLe | 75.1±0.4 | 61.1±0.2 | 49.2±1.2 | **70.4±0.6** |
| | PFR | **78.1±1.0** | 57.0±0.2 | 51.2±0.8 | 68.0±0.7 |
| | | UNSUPERVISED CONTINUAL LEARNING | | | |
| BarlowTwins | Finetune | 76.2±1.2 | 54.1±0.3 | 40.0±0.8 | 57.0±0.4 |
| | CaSSLe | **80.9±0.2** | **58.6±0.6** | 49.3±0.1 | **64.9±0.1** |
| | PFR | 78.8±0.6 | 57.2±0.2 | 46.0±0.7 | 61.1±0.2 |
| SimCLR | Finetune | 72.4±1.3 | 48.9±0.4 | 33.4±0.5 | 54.7±0.4 |
| | CaSSLe | 80.6±0.5 | 55.9±0.5 | 48.2±0.4 | 59.3±0.5 |
| | PFR | 79.2±0.7 | 53.8±0.3 | **49.4±0.1** | 57.7±0.2 |

Table 1: k-NN accuracy of the learnt representations. The best result in **bold** and second best underlined.

## 4.2 QUALITY OF REPRESENTATIONS

We investigate the quality of representations built by supervised and self-supervised training in a series of experiments. In Fig. 4 we show the accuracy of the k-NN classifier on CIFAR10 for all methods trained on different datasets and multiple combinations of two-step training. Overall, performance is the best when C10 is the final task and we do not have to address the problem of catastrophic forgetting. Slightly below is the performance of settings where C100 is the final task, as C10 and C100 are semantically similar. Training on SVHN, which is semantically more distant from C10, results in the worst representations to classify C10 classes and causes the biggest forgetting of useful features for all the methods (C10→SVHN). SL+MLP outperforms both self-supervised training and supervised training with linear head in all the experiments. SL+MLP is significantly better than the other methods when C10 is the first task to train on and the second task is C100 or SVHN.

**Forgetting** In Tab. 2 we observe high representation forgetting for SL, significantly lower for SSL, and the lowest for SL equipped with MLP projector.

**Task exclusion difference** In the two-task sequence $EXC$ answers the question: *what is the performance gap between the model trained on $B$ and a model trained on a sequence $A \rightarrow B$ when evaluated on $A$?* Results from Tab. 2 show that SL achieves small positive $EXC$ meaning that it forgets most features specific to the initial task (but not all of them). SL+MLP achieves the highest $EXC$ which shows that it is able to successfully retain a large portion of task-specific features. Surprisingly, SSL exhibits negative $EXC$. It means that it is better to train SSL model from scratch on another task than to finetune the model pretrained on the task of interest. In Fig. 3 we take a closer look at task exclusion difference presenting a training session for C10→SVHN in detail. We can see that only SL+MLP is able to retain a significant part of pretraining features resulting in a higher performance of the model trained on C10→SVHN than the model trained only on SVHN. We present more results in Appendix B.3.

**CKA similarity** In Tab. 2 we report CKA similarity between the models trained on C10 and the rest of the models. We observe that usage of MLP head in SL increases CKA between the C10 model

| Training | SL | | | | SSL | | | | SL+MLP | | | |
|---|---|---|---|---|---|---|---|---|---|---|---|---|
| sequence | $Acc_{C10}\uparrow$ | $F_{C10}\downarrow$ | $EXC_{C10}\uparrow$ | $CKA_{C10}\uparrow$ | $Acc_{C10}\uparrow$ | $F_{C10}\downarrow$ | $EXC_{C10}\uparrow$ | $CKA_{C10}\uparrow$ | $Acc_{C10}\uparrow$ | $F_{C10}\downarrow$ | $EXC_{C10}\uparrow$ | $CKA_{C10}\uparrow$ |
| C10 | 92.6±0.1 | - | - | - | 88.8±0.1 | - | - | - | **93.3±0.1** | - | - | - |
| C100 | 74.9±0.2 | - | - | 0.46±0.00 | 80.8±0.1 | - | - | **0.56±0.01** | 84.5±0.4 | - | - | 0.49±0.01 |
| C10→C100 | 76.1±0.1 | 16.6±0.2 | 1.2±0.3 | 0.50±0.00 | 79.1±0.2 | 9.7±0.3 | -1.8±0.2 | 0.52±0.01 | **88.8±0.2** | **4.5±0.3** | **4.3±0.6** | **0.57±0.00** |
| SVHN | 21.8±0.3 | - | - | 0.05±0.00 | **58.6±1.2** | - | - | **0.27±0.01** | 56.3±0.2 | - | - | 0.20±0.01 |
| C10→SVHN | 22.6±0.5 | 70.1±0.5 | 0.8±0.4 | 0.05±0.01 | 54.9±0.7 | 33.8±0.7 | -3.7±1.9 | **0.25±0.01** | **62.7±0.8** | **30.6±0.8** | **6.4±1.0** | **0.25±0.01** |

Table 2: We observe high representation forgetting for SL, significantly lower for SSL, and the lowest for SL equipped with MLP projector. SL is able to preserve a small fraction of task-specific features while SL+MLP can retain much more, based on their $EXC$ scores. Surprisingly, SSL achieves negative $EXC$ meaning that pretraining on a given task hurts the performance on this task after the finetuning. The best value between SL, SSL, and SL-MLP in **bold**.

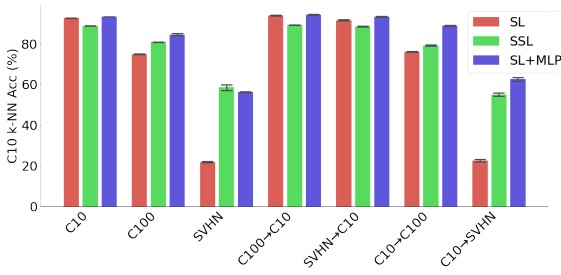

Figure 4: SL+MLP (blue) achieves better representations than SL (red) and SSL (green) regardless of the task sequence. We report task-aware k-NN performance on the CIFAR10 dataset (error bars in gray).

| Method | C10 | C100→C10 | | SVHN→C10 | |
|---|---|---|---|---|---|
| | $Acc_{C10}\uparrow$ | $Acc_{C10}\uparrow$ | $FT_{C10}\uparrow$ | $Acc_{C10}\uparrow$ | $FT_{C10}\uparrow$ |
| SL | 92.6±0.1 | 94.0±0.2 | **1.3±0.3** | 91.5±0.2 | -1.1±0.3 |
| SSL | 88.8±0.1 | 89.2±0.1 | 0.5±0.2 | 88.5±0.1 | -0.3±0.2 |
| SL+MLP | **93.3±0.1** | **94.3±0.1** | 1.0±0.0 | **93.2±0.2** | -0.1±0.1 |

Table 3: All methods benefit prom pretraining on C100 which is semantically close to C10. However, pretraining on semantically distant SVHN hinders the performance of SL.

and other models. Moreover, in the case of SL+MLP, the models pretrained on C10 and finetuned on another task have higher similarity to C10 models than the models trained on another dataset from scratch. This is not necessarily the case for SL models. SSL models have the highest CKA scores, however, they usually underperform compared to SL+MLP. This suggests that SSL produces similar features when trained on different datasets but their discriminative power for a classification task is worse than those learned with SL+MLP.

**Positive and negative forward transfer** We present the results of the forward transfer evaluation in Tab. 3. All the methods benefit from pretraining on CIFAR100 which is semantically close to CIFAR10. However, pretraining on semantically distant SVHN hinders the performance of SL but it hardly influences the performance of SSL and SL+MLP.

### 4.3 SPECTRA OF REPRESENTATIONS

To gain further insight into the properties of continually trained representations, we analyze the spectrum of their covariance matrix. We follow the procedure from Jing et al. (2022). We gather the representations of the validation set and compute the covariance matrix of the representations, $C$. We perform singular value decomposition of the covariance matrix $C = USV^T$, where $S = diag(\sigma^k)$ and $\sigma^k$ is $k$-th singular value of $C$. Fig. 5 presents how singu-

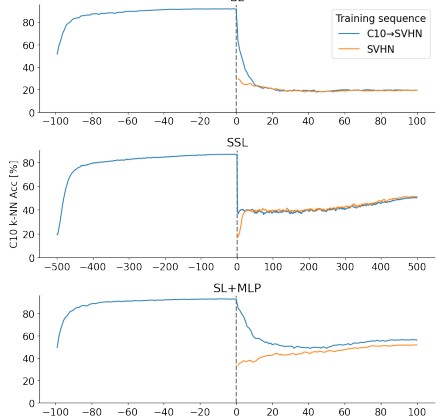

Figure 3: We compare the C10 accuracy of a model trained on sequence C10→SVHN (red) and a model trained from scratch on SVHN (green). SL (top) does not benefit from pretraining on C10 - the performance on C10 is almost the same for models trained C10→SVHN and on SVHN. Surprisingly, C10 pretraining harms SSL (middle) - the model trained from scratch on SVHN outperforms the model trained on C10→SVHN sequence when evaluating on C10. SL+MLP (bottom) is able to retain pretraining features resulting in higher performance of the model trained on C10→SVHN than the model trained only on SVHN.

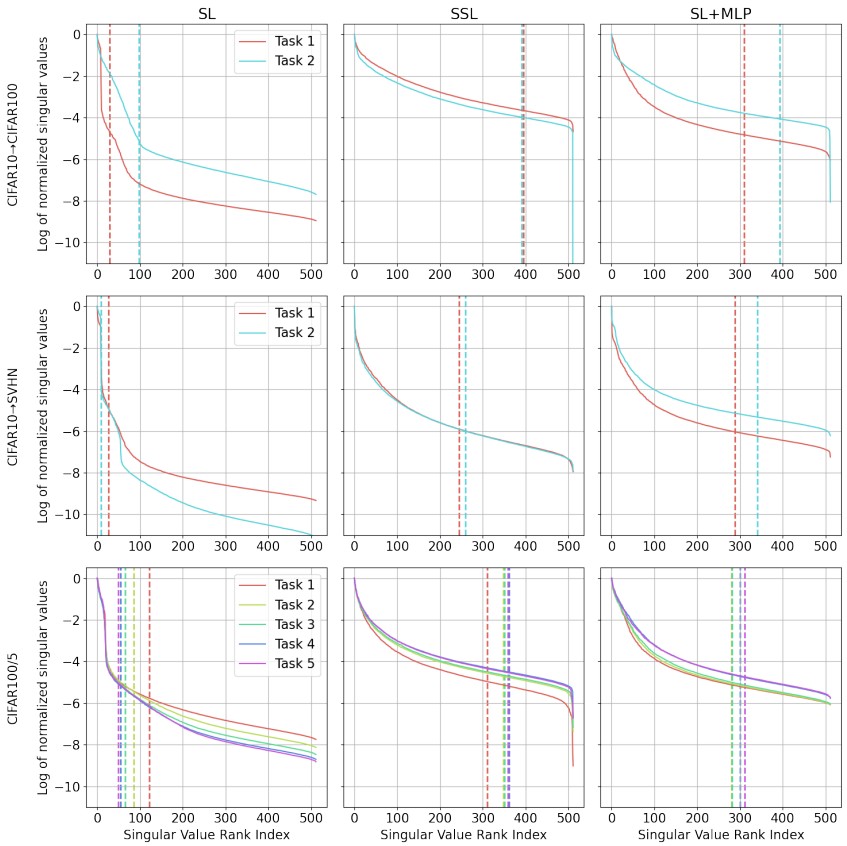

Figure 5: Singular value spectra of 512-dimensional representation space. Representations learned with SL+MLP (right) exhibit desirable properties from the continual learning point of view: (1) they consist of a more diverse set of features (contrary to SL, left); (2) they improve feature diversity when learning new tasks consistently across all the presented settings. Singular values are ordered descending and normalized and the scale is logarithmic. Vertical dashed lines denote 95% of the variance explained. Intuitively, it indicates how many relevant independent features the representation contains.

lar value spectra change after each task for different training methods and different sequences of tasks. Singular values are normalized by dividing them by $\sigma^1$ (the largest singular value).

**Representation collapse** Fig. 5 reveals that supervised learning exhibits signs of neural collapse (Papyan et al., 2020) - a large fraction of variance is described by a few dimensions roughly equal to the number of classes in the training set. This is an undesirable property in continual representation learning as the representations should be more versatile and useful not only for current but also for past and future tasks. SSL, on the other hand, learns a more diverse set of features resulting in a flatter singular values spectrum. In our experiments adding MLP to SL prevents neural collapse and results in features' properties more similar to SSL.

**Evolution of representations** An important property of representations learned in continual learning is the change in their diversity: the diversity that increases after each task is desired. In Fig. 5 we can observe that for SL, the diversity of the features usually decreases, except for C10→C100 where the increase is caused by a higher number of training classes (Papyan et al., 2020). For SSL, the diversity increases in the five-task scenario and remains close to constant for two-task settings. SL+MLP is able to improve the diversity of the representations consistently across all the presented scenarios suggesting its superiority in continual representation learning. It may be related to its ability to effectively accumulate knowledge when trained on a sequence of tasks, as presented in Fig. 2.

| | Str. | MLP | C10→C100 | | C100→C10 | | C10→SVHN | | SVHN→C10 | | C100/5 |
|---|---|---|---|---|---|---|---|---|---|---|---|
| | aug. | head | $Acc_{C10}$ ↑ | $Acc_{C100}$ ↑ | $Acc_{C100}$ ↑ | $Acc_{C10}$ ↑ | $Acc_{C10}$ ↑ | $Acc_{SVHN}$ ↑ | $Acc_{SVHN}$ ↑ | $Acc_{C10}$ ↑ | $Acc_{C100}$ ↑ |
| (1) | × | × | 76.1±0.1 | 74.4±0.0 | 28.2±0.9 | 94.0±0.2 | 22.6±0.5 | 95.9±0.2 | 27.7±0.9 | 91.5±0.2 | 38.5±0.4 |
| (2) | × | ✓ | 85.7±0.2 | 73.6±0.2 | 54.3±0.1 | 93.5±0.1 | 54.6±0.8 | 95.9±0.1 | **65.4±1.0** | 91.4±0.1 | 50.7±0.6 |
| (3) | ✓ | ✓ | **88.8±0.2** | **75.1±0.2** | **61.3±0.4** | **94.3±0.1** | **62.7±0.8** | **96.4±0.2** | 64.1±0.3 | **93.2±0.2** | **61.9±0.5** |

Table 4: Ablation study on the different components of supervised training. Using strong augmentations and MLP head comes with a trade-off: it improves the performance on the first task (transferability) but deteriorates the performance on the second task. The trade-off is much more beneficial for C100→C10 scenario. We report task-aware k-NN accuracy after training on the sequence of two tasks.

## 4.4 STABILITY OF REPRESENTATIONS

We define representations as *stable* when they do not drift in the representation space when the network is trained on a new task. The stability of representations is a desired property of SCL models as stable representations facilitate continual training of a classifier (Yu et al., 2020). On the other hand, UCL evaluation only measures the representations' strength and the relationship of stability and strength of representations is not obvious. One can imagine both stable and unstable representations can improve strength during continual training.

In this section, we evaluate the stability of representations of SL and SSL models. We use nearest mean classifier (NMC) accuracy to measure it in the context of SCL. After the first task, we calculate prototypes of each class as a mean feature of all the samples of this class. We evaluate the model and save the prototypes. Then, we train on the second task and evaluate the model using saved prototypes. We use the accuracy obtained by classification using old prototypes as a proxy of the stability of the representations. In the case of perfectly stable representations, both evaluations would result in the same accuracy while perfectly unstable representations would cause accuracy to drop to a random guess level. Moreover, we evaluate the updated model using prototypes recalculated on previous data (not allowed in continual learning) to provide an upper bound.

The results are presented in Fig. 6. Representations of all the methods are not stable in high distribution shift scenario C10→SVHN. They achieve random guess accuracy when utilizing saved (old) prototypes. However, in a low distribution shift scenario, C10→C100, SL achieves 53.1% accuracy using old prototypes (5.7% below upper bound performance) while SSL achieves 48.0% (13.5% below upperbound) and SL+MLP achieves only 36.7% (53.4% below upper-bound). Note that performance degradation can be only partially attributed to forgetting of representations as the upper-bound performance is still high after training on the second task for most of the methods. These results suggest that there exists a trade-off between the stability of representations and expressiveness of representations trained continually as methods that build stronger representations tend to have lower stability.

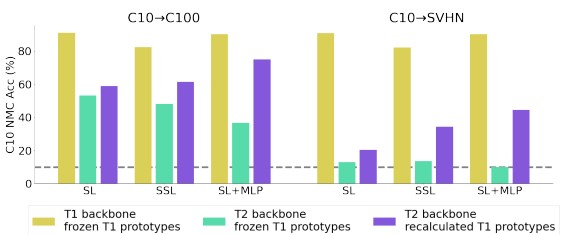

Figure 6: Task aware NMC accuracy on CIFAR10 dataset for supervised and self-supervised models trained on different sequences of tasks. After training on CIFAR10 (T1), both SL and SSL models achieve high NMC performance (yellow). After training the second task (T2), the nearest mean classification using old prototypes results in performance degradation (green). We calculate *an upper-bound* accuracy after training on the second task by recalculating the prototypes using old data and a new backbone (purple). Note that it is not possible in the CL scenario as old data is inaccessible. Gray dotted line marks random guess performance.

## 4.5 ABLATION STUDY

In Tab. 4 we inspect the contributions of the supervised training modifications. Specifically, we identify the importance of an MLP head and strong augmentations.

**MLP head** We use an additional MLP head during the supervised training as described in Wang et al. (2021). More specifically, MLP consists of a fully-connected layer, batch normalization layer, ReLU activation, and a fully-connected layer followed by a linear classification layer. The hidden feature dimension is set to 4096. The head is discarded after the training and the representations are probed from the ResNet backbone. In the following task, the MLP head is randomly reinitialized. Rows (1) and (2) show the effect of replacing a usual linear head with an MLP head. The MLP head consistently improves the performance on the first task and slightly diminishes the performance on the second task for each two-task sequence. However, the boost on the first task heavily surpasses the decline on the second task (average boost of 26.4% on the first task vs. average decline of 0.4% on the second task). The highly positive impact of the MLP head is also visible in the CIFAR100/5 scenario, boosting the performance by 12.2%. The impact of the projector's architecture is investigated further in Appendix B.4.

**Stronger augmentations** We refer to augmentations proposed in Wang et al. (2021) as *stronger augmentations*. We investigate whether strong augmentations facilitate supervised continual representation learning. Experiments that do not use stronger augmentations perform random resized crop with scale sampled uniformly from $[0.9, 1.0]$, random horizontal flipping, and input normalization. Rows (2) and (3) show the effect of stronger augmentations on the performance of trained representations. Stronger augmentations improve the performance in most scenarios. However, their impact is less significant than the impact of the MLP projector.

## 5 DISCUSSION AND LIMITATIONS

Although supervised learning with the MLP projection head seems to be more effective in continual representation learning, it comes at a price. SL requires mundane image labeling of the whole dataset which can be costly and impractical at scale. Self-supervised learning, on the other hand, is not dependent on image annotations and, therefore, can operate on a vast amount of unlabeled data.

However, SSL faces its own limitations. Firstly, most SSL approaches depend on strong image augmentations and learn representations that are invariant to them Chen et al. (2020); Zbontar et al. (2021); Caron et al. (2021). This can hinder the performance on the downstream tasks which require attention to the traits that it has been trained to be invariant to (Xiao et al., 2021). Moreover, SSL usually requires longer training which increases computational requirements in comparison to SL.

It is also worth noting that both SL+MLP and SSL introduce additional cost to the model during the training, as both introduce MLP projector that requires more computational requirements. However, at test time every method operates at the same number of parameters, as we discard MLP projectors after training.

Furthermore, it's worth re-emphasizing that this work focuses on continual representation learning. While we utilize data from previous tasks to construct k-NN and nearest mean classifiers for evaluating learned representations, our primary objective is not centered on the continual approach to the downstream task (classification). We are not delving into class-incremental learning, a prevalent continual learning setting. Nonetheless, our analysis of representation strength and stability can offer valuable insights into continual learning dynamics, potentially aiding in the creation of more effective algorithms for continual downstream task solutions.

## 6 CONCLUSIONS

In this work, we are first to show that supervised learning can significantly outperform self-supervised learning in continual representation learning. We achieve it by equipping SL with a simple MLP projector discarded after the training, following the common practice from SSL. We show that SL+MLP can be successfully coupled with several continual learning strategies, further improving the performance. Finally, we shed some light on the reasons for improved performance when using MLP with SL: better transferability, lower forgetting, and higher diversity of learnt features.

**Reproducibility Statement** Details needed to reproduce the results are provided in Section 3 and Appendix A.1. Moreover, we enclose the code repository in the supplementary material. It contains instructions and configuration files that allow the reproduction of all the experiments from the paper.

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

# A  Appendix

## A.1  Implementation details

### A.1.1  CL strategies

LwF (Li & Hoiem, 2018) is a classic SCL method for feature distillation. It distills the logits of the frozen network trained on previous tasks using cross-entropy loss. We pair it with SL methods that train with cross-entropy loss. We use the implementation from (Masana et al., 2023).

CaSSLe (Fini et al., 2022) is a method for self-supervised continual learning that utilizes a learnable MLP to project past features onto the new latent space for improved feature distillation. The distillation is performed on the outputs from the SSL projector with the loss function of a particular SSL method. Because of reliance on SSL-specific components, namely the projector and loss function, we do not pair CaSSLe with supervised approaches, except for SupCon which loss and architecture closely resemble SSL. We follow an official implementation of CaSSLe.

PFR Gomez-Villa et al. (2021) realizes a similar idea to CaSSLe. It also uses a learnable MLP projector to enhance feature distillation. However, it uses cosine similarity as a loss function and performs distillation on the outputs of the backbone network. Therefore, we pair it with both SL and SSL approaches as it does not rely on SSL-specific components. We present the chosen values of regularization hyperparameter $\lambda$ in Tab. 5. We selected the best $\lambda \in \{1, 3, 10, 15, 25\}$ separately for each method and dataset.

| Method | CIFAR10/5 | CIFAR100/5 | CIFAR100/20 | ImageNet100/5 |
|--------|-----------|------------|-------------|---------------|
| SL | 1.0 | 10.0 | 10.0 | 15.0 |
| SL+MLP | 3.0 | 3.0 | 10.0 | 1.0 |
| t-ReX | 3.0 | 3.0 | 10.0 | 1.0 |
| SupCon | 3.0 | 10.0 | 25.0 | 10.0 |
| BarlowTwins | 25.0 | 25.0 | 25.0 | 25.0 |
| SimCLR | 3.0 | 3.0 | 15.0 | 3.0 |

Table 5: PFR regularization hyperparameter $\lambda$ for different methods and datasets.

### A.1.2  k-NN evaluation

Each model is evaluated with a k-nearest neighbour classifier after training each task (offline evaluation). Moreover, we perform some experiments where we use k-nn evaluation after each epoch (online evaluation for Fig. 3 and Fig. 9).

For online evaluation, we perform extensive hyperparameter search and report results obtained by the best probe. We explore the following hyperparameters:

- $k \in \{5, 10, 20, 50, 100, 200\}$ - number of considered neighbours;
- distance function - we consider either euclidean distance or cosine similarity;
- temperature $T \in \{0.02, 0.05, 0.07, 0.1, 0.2, 0.5\}$ used only with cosine distance;

resulting in 42 k-NN probes per one offline evaluation.

For online evaluation, we use a fixed hyperparameter set: $k = 20$, cosine distance, and $T = 0.07$. This k-NN configuration often turns out to be one of the best in offline evaluation.

# B  Extended analysis

## B.1  Detailed two-task results

In Fig. 7 we present detailed results of two-task settings results summed up in Fig. 1. We can observe that self-supervised learning outperforms supervised learning on the first task while the opposite is true for the second task. SL equipped with MLP achieves the highest average accuracy on both tasks usually outperforming both SL and SSL on the first and second tasks.

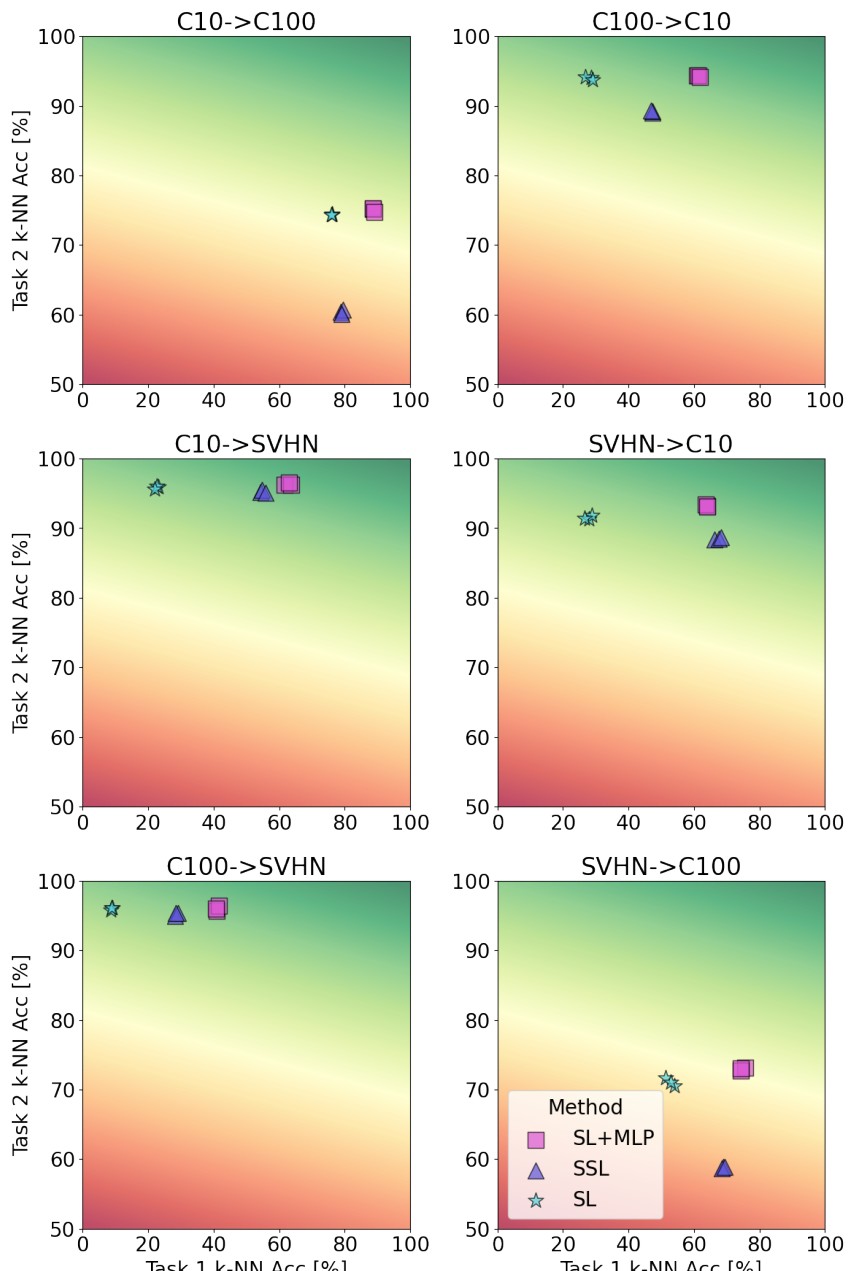

Figure 7: Results of two-task settings after training on the second task. Accuracy on the first task is presented on the horizontal axis and accuracy on the second task is presented on the vertical axis while the background color indicates the average accuracy on both tasks. SL usually outperforms SSL on the second task and usually underperforms on the first task. SL+MLP takes the best of both worlds (high first-task accuracy from SSL and high second-task accuracy from SL) and achieves the best overall performance.

## B.2 IMPACT OF TRAINING LENGTH

We investigate how the number of epochs influences the representations trained with different methods. We conducted experiments on a long sequence of tasks, C100/20, training with SSL, SL, and SL+MLP methods for a given number of epochs in each task. We present the results in Tab. 8.

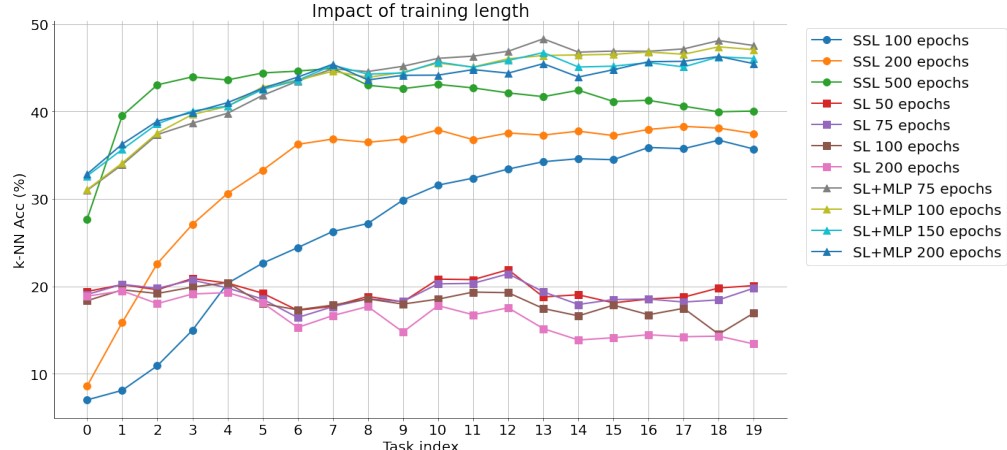

Figure 8: Self-supervised models benefit from longer training. However, supervised models, both with and without MLP projector result in reduced performance when trained for a large number of epochs. We report task-agnostic k-NN accuracy for all tasks after each task.

| Method | Projector arch. | CIFAR10/5 | CIFAR100/5 |
|--------|-----------------|-----------|------------|
| SL | None | 56.9±1.4 | 38.5±0.4 |
| | MLPP | 65.9±0.7 | **61.9±0.5** |
| | t-ReX | **67.3±0.1** | 58.3±0.2 |
| SupCon | Base | 60.4±0.6 | 49.4±0.3 |
| | MLPP | **66.2±0.7** | 54.3±0.5 |
| | t-ReX | 63.5±0.2 | **58.1±0.1** |
| SimCLR | Base | 72.4±1.3 | 48.9±0.4 |
| | MLPP | 76.1±0.8 | 52.9±0.4 |
| | t-ReX | **79.6±0.2** | **56.3±0.1** |

Table 6: Impact of projector architecture on different methods. In most cases, the bigger the projector the better the performance. Best results for each method in **bold**.

### B.3 Task exclusion comparison

In Fig. 9 we take e closer look at the task exclusion comparison. We identify that the training recipe is a factor responsible for its negative task exclusion difference. The training recipe for SL and SSL differs: SL is trained for 100 epochs with a 0.025 learning rate while SSL is trained for 500 epochs with 0.3 learning rate. When training SSL for 100 epochs with a learning rate of 0.025, following the supervised learning recipe, we observe that SSL exhibits positive behavior that is similar to SL+MLP. However, such training configuration leads to the suboptimal final performance of a continual learner, as shown in Fig. 8.

### B.4 Impact of projector architecture

We investigate the impact of various architectures for MLP projector presented in Fig. 10. We evaluate the projector proposed by SimCLR (Chen et al., 2020), t-ReX (Sariyildiz et al., 2023) and by Wang et al. (2021). We present the results of evaluated architectures coupled with SL, SupCon, and SimCLR in Fig.10.

### B.5 Extended ablation study

In this section, we extend the ablation study presented in Sec. 4.5. We present singular value spectra of all three configurations for four sequences in Fig. 11.

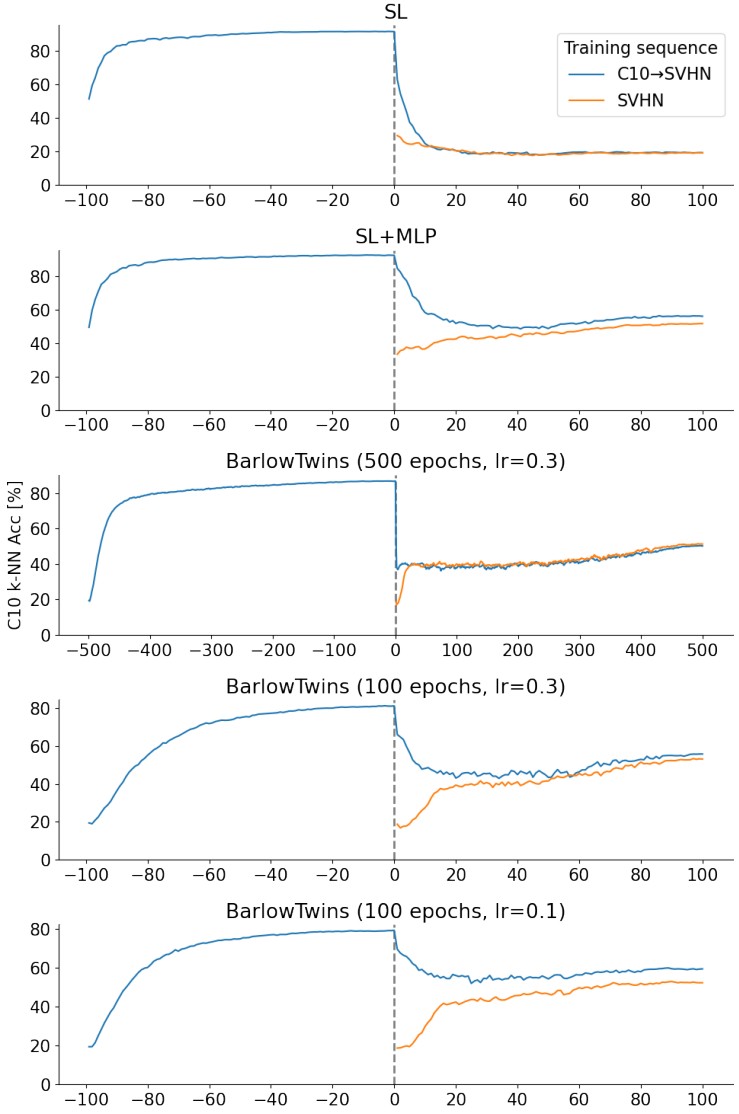

Figure 9: SSL behaves similarly to SL+MLP when trained for the same number of epochs with the same learning rate.

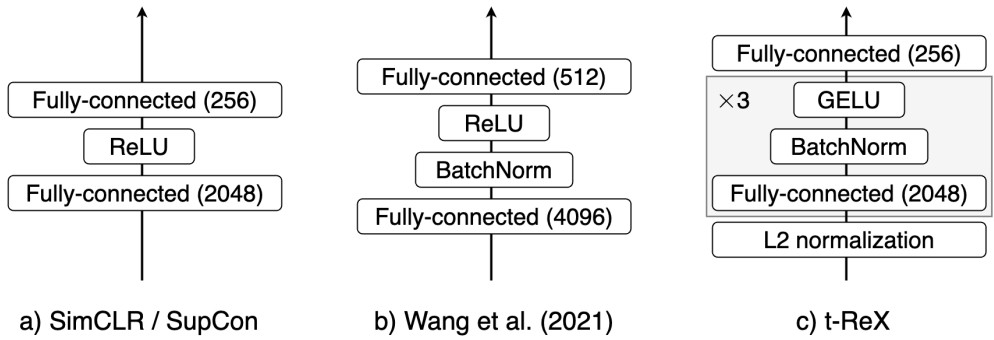

Figure 10: Architectures of the projectors used by different methods.

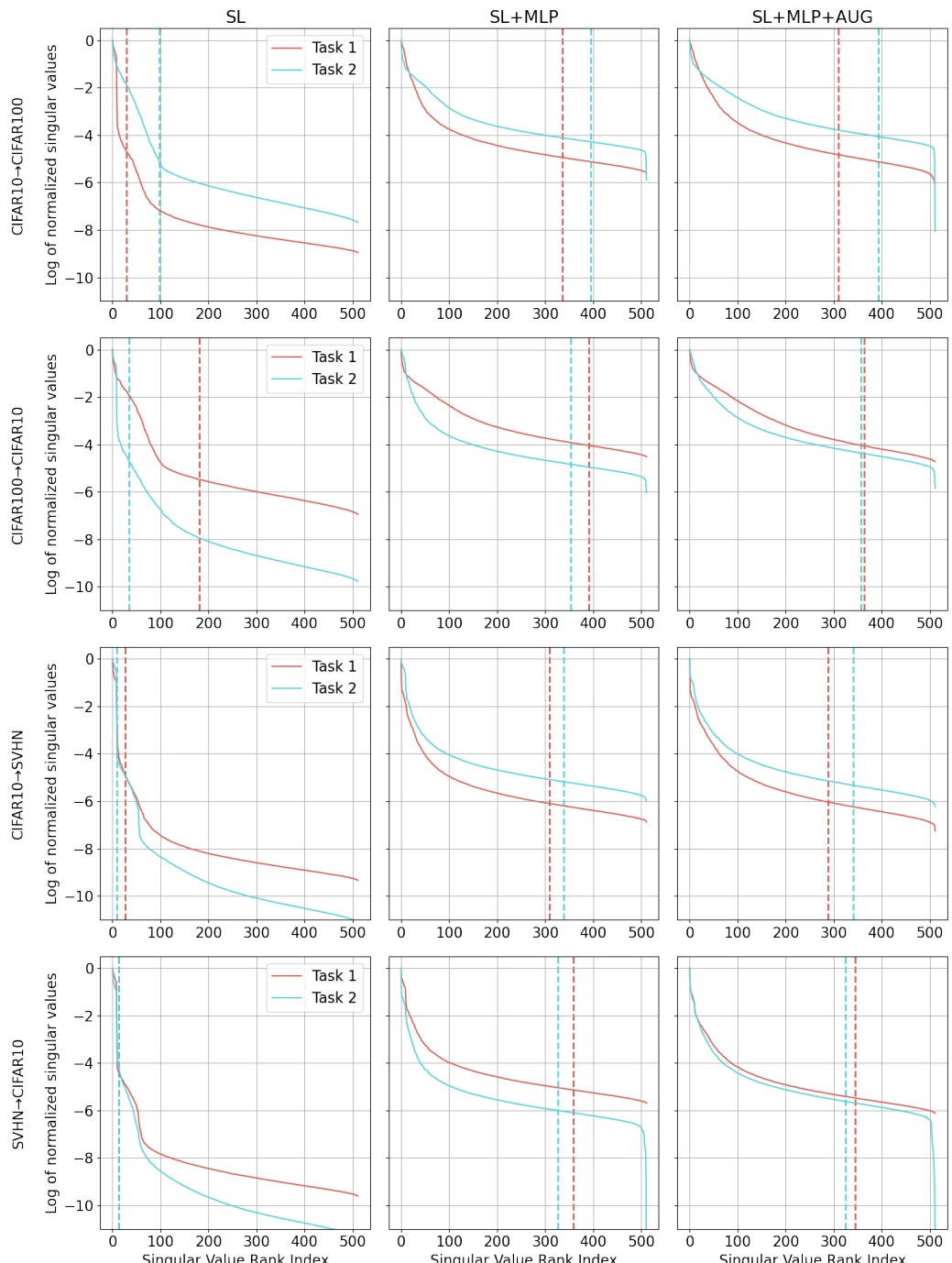

Figure 11: Singular value spectra of 512-dimensional representation space. SL (left) uses weak augmentations and linear head, SL+MLP (middle) employs MLP projector and SL+MLP+AUG (right) uses stronger augmentations during training. We can observe that the use of MLP projector leads to flatter spectra (significant difference between left and middle). However, the use of stronger augmentations has a marginal impact on representation spectra (no significant difference between middle and right). Note that "weak/stronger augmentations" relate to augmentations used during training and all approaches use the same augmentations to produce representations for evaluation, e.g. this spectral analysis.

