# OpenReview forum: "Revisiting Supervision for Continual Representation Learning"
_ICLR.cc/2024/Conference — ICLR 2024 Conference Withdrawn Submission_

### Official Review · Reviewer_Zari · 2023-10-26

**Soundness:** 2 fair
**Presentation:** 3 good
**Contribution:** 2 fair
**Rating:** 3
**Confidence:** 5

**Summary:**

The paper reexamines the quality of representations produced by supervised continual learning (CL) methods, operating under the premise that the presence of labels should not lead to a degradation of representation quality. In exploring this, the paper draws from findings in the transfer learning domain, pinpointing the MLP projector as a crucial element for enhancing the performance of supervised learning techniques. The paper experiments on an array of standard continual learning (CL) datasets such as sequential CIFAR-10, CIFAR-100, SVHN and ImageNet100. Across these datasets, the findings consistently highlight the tangible enhancements brought about by integrating the MLP projector with various CL methodologies.

**Strengths:**

- The paper presents an insightful exploration into the integration and impact of an MLP projector in supervised continual learning (SCL). One of the key findings is the significant improvement in fine-tuning performance across various datasets, achieved by simply adding the MLP projector. This aspect of the study is particularly noteworthy, as it suggests a relatively straightforward method to enhance existing SCL approaches.
- The analysis conducted on the influence of the projector within SCL, including aspects like representation collapse and the evolution of representations, offers valuable insights. These findings could be valuable for the research community in developing new methods in future.
- The paper is overall clear, well-written, and easy to understand.

- The paper's supplementary material contains the code, which can facilitate its adoption in future comparisons between SCL, SCL + MLP and UCL methods.

**Weaknesses:**

- **Motivational clarity.** I'd like to request more details on the motivation behind this study. Although the investigation into why annotations negatively impact performance in supervised settings is intriguing, it's important to recognize that in real-world applications, access to a complete set of labels for new tasks is rarely possible.  It could be beneficial to explore how the performance of supervised methods evolves as the proportion of available labels increases. Such an analysis might offer valuable insights into the point at which SCL + MLP begins to outperform UCL methods. This comparative perspective could greatly enhance the practical relevance of the paper.
- **Originality in Methodology.** While the paper's integration of the projector into the SCL framework is notable, it primarily seems to extend findings from Wang et al., 2021, and Sariyildiz et al., 2023, in the context of continual learning. These prior studies have similarly explored MLP's role in enhancing supervised learning methods' generalizability and transferability.
- **Scope of Experimental Evaluation:** Third, the experimental evaluation in the paper appears somewhat limited. By focusing solely on the continual fine-tuning scenario, the scope of the insights offered by the paper is somewhat restricted. The baselines selected for comparison in both supervised and unsupervised continual learning contexts are somewhat dated and not particularly strong. Notably, there's an absence of comparisons with more recent supervised methods such as DER, DER++, GMED (Jin et al., 2021), and CLS-ER (Arani et al., 2022). Moreover, the paper does not include a comparison with LUMP (Madaan et al., 2022), which could be a significant omission given the context of the research. Including these comparisons could potentially strengthen the paper’s empirical evaluation and its position within the current research landscape.
- **Theoretical understanding.** While the paper sheds light on representation collapse and CKA similarity, justifying MLP's use in SCL, a more detailed analysis or theoretical explanation of these phenomena in this specific context would be beneficial. Moreover, a thorough theoretical rationale for why combining SSL with MLP is particularly advantageous for continual learning could substantially enrich the paper.
- **Additional Suggestions.**
   * It would be useful to include only the strong augmentations in Table 4.
   - In Figure 8, employing distinct line styles for various methods could greatly improve the graph's interpretability.
   - Section B.3, "Take e closer" -> "Take a closer"

**Questions:**

- Could the authors provide more details on the method used to measure diversity in the representations on page 7? Specifically, how is the assertion substantiated that feature diversity decreases in Supervised Learning (SL) but increases in Self-Supervised Learning (SSL)?
- It's noted that SSL exhibits less forgetting compared to SL, yet the EXC metric is higher for SSL. Comment.
- In some instances, Finetuning seems to outperform Learning Without Forgetting (LWF) in the context of SL combined with a Multi-Layer Perceptron (MLP). Comment.
- The paper suggests that random reinitialization of the MLP head after each task may contribute to the enhanced performance seen in SCL (Supervised Contrastive Learning) combined with MLP. To ascertain whether the improvement is due to the MLP itself or its reinitialization, it would be essential to include the following ablation experiments:
   - SCL + MLP where the MLP is not reinitialized after each task.
   - Reinitializing the MLP projector in Unsupervised Contrastive Learning (UCL) after every task.
- Regarding Figure 8, is the k-NN accuracy measured on the validation set? Also, was the best-performing model used for the main results reporting?
- For Figure 9, could the authors expand on the caption and explain the experimental setup, particularly how SSL parallels SL + MLP when both are trained for an equivalent number of epochs with the same learning rate?

---

### Official Review · Reviewer_6fut · 2023-10-31

**Soundness:** 2 fair
**Presentation:** 3 good
**Contribution:** 2 fair
**Rating:** 3
**Confidence:** 3

**Summary:**

This paper presents an experimental analysis of combining supervised learning with a MLP in the continual representation learning problem. The paper aims at showing that this strategy can help overcome forgetting, and therefore learn tasks continually more effectively. Experiments are run using supervised learning with MLP in combination with some SOTA continual learning methods, over benchmark continual learning datasets. The quality of the obtained representations is measured using kNN accuracy, and also forgetting, forward and backward transfer, and stability of representations.

**Strengths:**

- The paper addresses an important problem of learning good representations in the continual learning setting, and of comparing widely-adopted strategies including supervised learning and self-supervised learning.
- The paper is well written. The experiments, especially, are well-organised and presented.

**Weaknesses:**

- Apart from a very general description in Figure 1, there are not many technical details of the proposed method, and no insights of the inner-workings. Although I appreciate the comprehensiveness of the experimental, I fail to see the contribution of the paper beyond a mere experimental analysis where apparently a well-established method can do very well (SupCon in Table 1).
- In the experiments, there is no reasoning around the SOTA supervised continual learning methods that were selected for combination with MLP. Considering the wide variety of these approaches (including regularization-based, knowledge distillation-based, replay-based and network expansion-based methods), why were these three SL methods selected, and no others? Also, three methods seems quite a limited number of SOTA methods to actually be convinced that MLP can be effectively combined with any existing SL method.
- In the experiments, I failed to see if you were doing any modification to SupCon at all? If you are not, and since this method is quite competitive according to Table 1, what is the contribution?

**Questions:**

Please refer to "weaknesses" section.

---

### Official Review · Reviewer_vMLn · 2023-10-31

**Soundness:** 3 good
**Presentation:** 2 fair
**Contribution:** 2 fair
**Rating:** 5
**Confidence:** 2

**Summary:**

Continual learning involves cumulative learning, or learning more and more, as one moves from one task to another, and
should involve improving one's representations (eg generality/transferability) and not forgetting the important stuff learned before, such as not forgetting how to detect the previous classes (previous ml tasks) one has learned to perform well on. In the context of several vision
datasets (classification of images), the authors provide evidence that supervision, as long as the networks have a multi-layer perceptron projector (MLP) added on, could do just as well and often better than the self-supervised techniques, in terms of the quality of the representations learned (when evaluated via nearest neighbors techniques using the learned representations). Recent research had suggested the self-supervised learning could be better, which is puzzling (how can extra relevant label information degrade performance), but this paper (along with other recent work) provides evidence for the importance of  the MLP layer in these findings.

**Strengths:**

Extensive empirical experiments and comparisons to help clarify a puzzling recent empirical finding: The authors convincingly empirically show the importance or utility of MLP projection (both for SL and SSL) using several datasets and image classification task sequences and several train model techniques.  It is good to provide explanations for the somewhat puzzling recent findings.

**Weaknesses:**

I am uncertain whether the broader research approach or the problem setting has  sufficient generality (the audience for this work may be too limited), so I am borderline: how are the learned representations used in practice, in some practical classification task? The applications (use of the learned representations via some nearest neighbor technique) appears highly indirect currently. Also: doesn't SSL (or unsupervised continual learning) have
the potential advantage of access to much more data, as the authors note, for example in section 5 discussions (eg orders of magnitude) (putting aside computational cost concerns). I don't think the authors evaluate under the scenario of one or more orders of magnitude data available for the unsupervised techniques. Perhaps this is not the goal of this work, but this aspect, limits the utility of this work.

Finally, the importance of MLP projection for SL had already been shown in somewhat similar settings (as the authors cite: Sariyildiz et
al ICLR2023 and Wang et al, 2021, improving transferability of supervised learning via MLP projectors), rendering the contribution
of this work incremental.

**Questions:**

Please emphasize that the comparisons are done with exact same datasets, and in particular with the same training sizes. What would
happen if the available unlabeled data becomes much more plentiful?

Other minor:

- introduction: replace founding' with 'finding', in "Following that founding Wang et al. (2021);..."

-  page 2, section 2: if there are no annotations what does 'positives' mean: 'by ensuring that the network generates similar
 representations for the positives' (do you mean positives that are unknown to the techniques? needs clarification)

-  pg 3, citation not found in: (?Yan et al., 2021)